# ON LEARNING WITH FAIRNESS TRADE-OFFS

## ABSTRACT

Previous literature has shown that bias mitigating algorithms were sometimes prone to overfitting and had poor out-of-sample generalisation. This paper is first and foremost concerned with establishing a mathematical framework to tackle the specific issue of generalisation. Throughout this work, we consider fairness trade-offs and objectives mixing statistical loss over the whole sample and fairness penalties on categories (which could stem from different values of protected attributes), encompassing partial de-biasing. We do so by adopting two different but complementary viewpoints: first, we consider a PAC-type setup and derive probabilistic upper bounds involving sample-only information; second, we leverage an asymptotic framework to derive a closed-form limiting distribution for the difference between the empirical trade-off and the true trade-off. While these results provide guarantees for learning fairness metrics across categories, they also point out to the key (but asymmetric) role played by class imbalance. To summarise, learning fairness without having access to enough category-level samples is hard, and a simple numerical experiment shows that it can lead to spurious results.

## 1 INTRODUCTION

Reducing bias in an algorithm requires a number of steps; first, specify the fairness definitions (and thus fairness metrics) that apply, second, encode them with a penalty so as to measure the discrepancy between outcomes and the perfect fairness scenario, third, choose the trade-off between the original statistical goal and fairness constraints, and, fourth, pick a method to debias (at least partially) the model. There are a number of obstacles to this programme. Indeed, debiasing may come at a cost (Rodolfa et al. (2020)), but there are also theoretical reasons behind this claim. Jointly tackling multiple fairness definitions is usually difficult if not impossible without a decrease in model performance due to a series of impossibility theorems (Kleinberg et al. (2017); Pleiss et al. (2017); Chouldechova (2017); Kim et al. (2020)). Note that these results apply not only to fairness-performance but also to fairness-fairness trade-offs.

Research has recently been undertaken to tackle specifically learnability and generalisation in fairness-related problems. In particular, Oneto et al. (2020); Oneto et al. (2020) derive provable probabilistic upper bounds in a fairness setting where one estimates an overall statistical loss and monitors disparities across categories. In Chen et al. (2018), the authors have applied bias-variance decomposition techniques to disentangle the sources of unfairness in a classifier. While this is not directly related to this present work, the principle of risk decomposability turns out to be fruitful in both setups. Finally, in Agrawal et al. (2020), a limiting distribution is established in the simple case where there is a partial requirement involving a fairness metric with two categories.

**Our contributions.** The novelty of our work consists of the application of known techniques such as PAC inequalities and the central limit theorem to the problem of learning under fairness constraints:

- First, we develop a PAC framework for fairness trade-offs, generalising results from Donini et al. (2018); Oneto et al. (2020) and tackling explicitly *partial debiasing* (see Kim et al. (2020); Agrawal et al. (2020)).
- Second, we show that certain properties of the probabilistic upper bounds lead to the need for sample-efficient bias mitigation techniques. In particular, we show that a new quantity, $Z_S$, which can be understood as measure of sample concentration, plays a crucial role. We

also build on Agrawal et al. (2020) and develop an asymptotic framework with a known limiting distribution; this is useful as PAC bounds may not always be very sharp.

- Third, in both frameworks, we put forward the decomposability of the learning risk, expressed as an upper bound in the PAC realm and a limiting variance in the asymptotic one.

- Last, we illustrate our results on real-life data and show that generalisation is indeed difficult.

## 2 FAIRNESS METRICS AND LOSS FUNCTIONS

### 2.1 SET-UP AND DEFINITIONS

In this article, we define $s = 1, \cdots, C$ to be a categorical (protected) attribute, $\mathbf{x} \in \mathcal{X} \subset \mathbb{R}^d$ to be a set of non-protected features excluding $s$. $y \in \{-1, +1\}$ is a binary outcome variable and $\hat{y} \in \{-1, +1\}$ is an estimator for $y$. Finally, $z = (\mathbf{x}, y, s)$. Note that $\hat{y}$ is derived from a learner $h \in \mathcal{H}$, where $\mathcal{H}$ is a given functional space. Furthermore, we define $S$ to be the in-sample (training) empirical distribution, and $D$ to refer to the true distribution.

Throughout this paper, we consider the case of an overall objective as a statistical performance objective, $L^0$, plus a fairness loss function $\phi$. The trade-off is tuned by a hyper-parameter $\lambda \geq 0$. As usual, the aim is to minimise the overall objective, i.e., minimise the statistical loss and the lack of fairness. This set-up is typical for partial debiasing and can be found in Kim et al. (2020).

**Definition 1.** The *overall objective* or *fairness trade-off* is defined as

$$
\begin{aligned}
\mathcal{L}_D(h) &= L_D^0(h) + \lambda\phi\left(L_{D,1}^+(h), L_{D,1}^-(h), \cdots, L_{D,C}^+(h), L_{D,C}^-(h)\right) \\
&= L_D^0(h) + \lambda\phi(\mathbf{L}_D^{\pm}(h)),
\end{aligned}
$$

where we have used the standard notations (see Shalev-Shwartz & Ben-David (2014)):

$$
\begin{aligned}
L_D^0(h) &= \mathbb{E}\left[\ell^0(h, z)\right] \\
L_{D,a}^+(h) &= \mathbb{E}\left[\ell^+(h, z)|s = a, y = 1\right] \\
L_{D,a}^-(h) &= \mathbb{E}\left[\ell^-(h, z)|s = a, y = -1\right]
\end{aligned}
$$

and the vector notation $\mathbf{L}_D^{\pm}(h) = \left[L_{D,1}^+(h), L_{D,1}^-(h), \cdots, L_{D,C}^+(h), L_{D,C}^-(h)\right]^T$, for some functions $\ell^0$, $\ell^+$ and $\ell^-$.

*Remark* 1. In the whole paper, we assume that there exists a uniform bound $B > 0$ on all functions $\ell$, i.e., $|\ell(h, z)| \leq B$ for all $h$ and $z$. $\ell$ refers to any such function in what follows.

Let us detail two particular cases. If $\lambda = 0$, then the overall objective boils down to the usual risk minimisation problem. If $\ell^0 = 0$ or $\lambda \to +\infty$, then the trade-off becomes a fairness constraint.

We are now in a position to define the empirical counterpart by considering the empirical distribution rather than the true distribution. With the additional notations $\mathcal{N}_a^{\pm} = \{i \in \{1, \cdots, n\}; s_i = a, y_i = \pm 1\}$, $|\mathcal{N}_a^{\pm}| = n_a^{\pm}$, the corresponding *sample* versions can be defined as $L_S^0(h) = \frac{1}{n}\sum_{i=1}^n \ell^0(h, z_i)$, $L_{S,a}^+(h) = \frac{1}{n_a^+}\sum_{i \in \mathcal{N}_a^+} \ell^+(h, z_i)$ and $L_{S,a}^-(h) = \frac{1}{n_a^-}\sum_{i \in \mathcal{N}_a^-} \ell^-(h, z_i)$, where $n_a = n_a^+ + n_a^-$, $n = \sum_{a=1}^C n_a$, leading to

$$
\mathcal{L}_S(h) = L_S^0(h) + \lambda\phi\left(\mathbf{L}_S^{\pm}(h)\right) \tag{1}
$$

For the sake of simplicity, we will generally refer to $\phi(\mathbf{L}_T^{\pm}(h))$ as $\phi_T(h)$ for $T = D, S$.

### 2.2 FAIRNESS DEFINITIONS AND METRICS

One can find multiple technical definitions of fairness in the literature; they have been reviewed in various papers (Narayanan, 2018; Verma & Rubin, 2018; Berk et al., 2018; Kim et al., 2020; Agrawal et al., 2020). We offer an overview of the most frequent metrics in Table 1.

We note that all fairness metrics considered here can be expressed as an equality requirement on probabilities, hence on the expectation of indicator variables. These fall within our framework. However, this framework can also handle other types of functions.

| Fairness metric | Reference | Equality requirement |
|---|---|---|
| Equalized false omission rate | (Berk et al., 2018) | $\mathbb{P}(y = 1|\hat{y} = -1, s = a) = \mathbb{P}(y = 1|\hat{y} = -1)$ |
| Predictive parity | (Chouldechova, 2017) | $\mathbb{P}(y = 1|\hat{y} = 1, s = a) = \mathbb{P}(y = 1|\hat{y} = 1)$ |
| Demographic parity | (Calders & Verwer, 2010) | $\mathbb{P}(\hat{y} = 1|s = a) = \mathbb{P}(\hat{y} = 1)$ |
| Equalized false negative rate | (Chouldechova, 2017) | $\mathbb{P}(\hat{y} = -1|y = 1, s = a) = \mathbb{P}(\hat{y} = -1|y = 1)$ |
| Predictive equality | (Chouldechova, 2017) | $\mathbb{P}(\hat{y} = 1|y = -1, s = a) = \mathbb{P}(\hat{y} = 1|y = -1)$ |
| Equality of opportunity | (Hardt et al., 2016) | $\mathbb{P}(\hat{y} = 1|y = 1, s = a) = \mathbb{P}(\hat{y} = 1|y = 1)$ |
| Equalized odds | (Hardt et al., 2016) | $\mathbb{P}(\hat{y} = 1|y = y', s = a) = \mathbb{P}(\hat{y} = 1|y = y')$ |

Table 1: Frequent fairness definitions.

## 2.3 FAIRNESS LOSS FUNCTIONS

In addition to picking one or multiple fairness definitions, we need to specify $\phi$ to measure the discrepancy from perfect fairness, the ideal case.

A first approach, similar to the Calders-Verwer gap (Calders & Verwer, 2010), consists of looking at the discrepancy between two categories $a$ and $a'$:

$$\Delta_{T,a,a'}^{L,\pm}(h) = L_{T,a}^{\pm}(h) - L_{T,a'}^{\pm}(h),$$

for $T \in \{D, S\}$. It is worth 0 under perfect fairness, but is asymmetric and thus requires defining an advantaged (or benchmark) category $a$.

To avoid the issue of asymmetry, one can follow Oneto et al. (2020), and define $\phi$ as the sum of all absolute discrepancies across categories:

$$\phi(\mathbf{L}_T^{\pm}(h)) = \sum_{a \neq a'} \left\{ \left| L_{T,a}^+ - L_{T,a'}^+ \right| + \left| L_{T,a}^- - L_{T,a'}^- \right| \right\}, \tag{2}$$

for $T = D, S$. Notice that, thanks to the reverse triangle inequality, the function $\phi$ is Lipschitz continuous. One could naturally weigh the various contributions to this fairness function and use another norm than $L^1$. Other functions involving ratios or relative differences, are also possible.

Finally, in Kim et al. (2020), the authors consider a convex combination of multiple loss functions $\phi^{(1)}, \cdots, \phi^{(M)}$:

$$\phi(\mathbf{L}_T^{\pm}(h)) = \sum_{j=1}^{M} \lambda_j \phi^{(j)} \left( \mathbf{L}_T^{\pm}(h) \right).$$

Note that if the $\phi^{(j)}$'s are Lipschitz-continuous (with respect to the loss vector $\mathbf{L}_T^{\pm}(h)$), with respective Lipschitz constant $K^{(j)}$, then the overall loss $\phi$ is itself Lipschitz-continuous with constant $K^\phi = \sum_{j=1}^{M} \lambda_j K^{(j)}$.

## 3 LEARNING FAIRNESS TRADE-OFFS

In this section, we consider the learning problem from different angles. We start by considering the statistical loss $L^0$ and derive a bound based on the Rademacher complexity observed in each category. To do so, we borrow from the statistical learning toolbox and briefly review Rademacher complexities. We then move on to learning the fairness part per se, namely study the generalisation properties of $\phi_S(h)$. Finally, we bring the two together and derive probabilistic upper bounds on fairness trade-off generalisation.

## 3.1 LEARNING THE STATISTICAL LOSS

Let us first focus on the *statistical* component of the loss, i.e., $L_T^0$ for $T \in \{S, D\}$. Much of this paper is based on theory of bounds derived from Rademacher complexities, introduced in Bartlett & Mendelson (2003) and surveyed in Boucheron, Stéphane et al. (2005). Recent textbooks such as Shalev-Shwartz & Ben-David (2014); Mohri et al. (2012) provide excellent introductions to the topic. We start by recalling the definition of Rademacher complexities, as indicated in Boucheron, Stéphane et al. (2005):

**Definition 2.** The Rademacher complexity (or average) of a function $\ell$ is given by

$$R(\ell \circ \mathcal{H} \circ \mathcal{S}) = \mathbf{E}\left[\sup_{h \in \mathcal{H}} \frac{1}{n}\left|\sum_{i=1}^{n} \sigma_i \ell(h, z_i)\right|\right]. \tag{3}$$

Note that definitions can slightly vary (depending for instance on the presence or absence of the absolute value in the definition), but all downstream results are qualitatively similar, up to some multiplicative constants.

To make this more concrete, we can consider a simple (but usual) case (as describe in Shalev-Shwartz & Ben-David (2014)). We suppose that almost surely $\|\mathbf{x}\|_2 \leq R$. In addition, if we let $\mathcal{H} = \{\mathbf{w}; \|\mathbf{w}\|_2 \leq R'\}$ and assume that the loss function $\ell$ is of the type $\ell(\mathbf{w}, (\mathbf{x}, y)) = \rho\left(\mathbf{w}^T \mathbf{x}, y\right)$, where $|\rho|$ is bounded by $B$ and is $L^\rho$-Lipschitz, then, almost surely,

$$R(\ell \circ \mathcal{H} \circ \mathcal{S}) \leq \frac{L^\rho R R'}{\sqrt{n}}.$$

Rademacher complexities are key to establishing learnability bounds and lead to fundamental results in statistical learning theory. In particular, we will make use of a standard result (see Boucheron, Stéphane et al. (2005)).

**Proposition 1.** *With probability at least $1 - \delta$, it holds*

$$\sup_{h \in \mathcal{H}} |L_D(h) - L_S(h)| \leq 2R(\ell \circ \mathcal{H} \circ \mathcal{S}) + B\sqrt{\frac{2\log\frac{2}{\delta}}{n}}. \tag{4}$$

It will also be useful to introduce *conditional* Rademacher complexities that will be used throughout this paper. In particular, since we have a partition of the sample in terms of categories $\mathcal{S} = \bigcup_{a=1}^{C} \mathcal{N}_a$, we can consider the Rademacher complexity of that particular sample:

$$R(\ell \circ \mathcal{H} \circ \mathcal{N}_a) = \mathbf{E}\left[\sup_{h \in \mathcal{H}} \frac{1}{n_a}\left|\sum_{i \in \mathcal{N}_a} \sigma_i \ell(h, z_i)\right|\right]. \tag{5}$$

**Lemma 1.** *The sample Rademacher complexity can be bounded from above by the weighted sum of conditional Rademacher complexities:*

$$R(\ell \circ \mathcal{H} \circ \mathcal{S}) \leq \sum_{a=1}^{C} \frac{n_a}{n} R(\ell \circ \mathcal{H} \circ \mathcal{N}_a), \tag{6}$$

*where $R(\ell \circ \mathcal{H} \circ \mathcal{N}_a) = \mathbf{E}\left[\sup_{h \in \mathcal{H}} \frac{1}{n_a}\left|\sum_{i \in \mathcal{N}_a} \sigma_i \ell(h, z_i)\right|\right]$.*

*Proof.* This comes directly from the sub-additivity of the absolute value and the supremum. □

One can further interpret the non-negative gap $\sum_{a=1}^{C} \frac{n_a}{n} R(\ell \circ \mathcal{H} \circ \mathcal{N}_a) - R(\ell \circ \mathcal{H} \circ \mathcal{S})$ as a diversification benefit amongst categories. Finally, this leads us to a proposition leveraging our results so far:

**Proposition 2.** *With probability at least $1 - \delta$,*

$$\sup_{h \in \mathcal{H}} \left|L_D^0(h) - L_S^0(h)\right| \leq 2 \sum_{a=1}^{C} \frac{n_a}{n} R(\ell^0 \circ \mathcal{H} \circ \mathcal{N}_a) + B\sqrt{\frac{2\log\frac{2}{\delta}}{n}}. \tag{7}$$

### 3.2 LEARNING FAIRNESS REQUIREMENTS

Let us now move on to the *fairness* learning part; as before, we would like to find a bound on the *distribution* fairness loss given the *sample* fairness loss, i.e., a probabilistic bound on the difference $\phi_D(h) - \phi_S(h)$.

**Proposition 3.** *Under the assumption that $\phi$ is $K^\phi$-Lipschitz, it holds, with probability at least $1-\delta$, and where $Z_S := \sum_{a=1}^{C} \sqrt{\frac{n}{n_a^+}} + \sqrt{\frac{n}{n_a^-}}$, that*

$$
\sup_{h \in \mathcal{H}} |\phi_D(h) - \phi_S(h)| \leq 2K^\phi \sum_{a=1}^{C} R(\ell^+ \circ \mathcal{H} \circ \mathcal{N}_a^+) + R(\ell^- \circ \mathcal{H} \circ \mathcal{N}_a^-)
$$

$$
+ K^\phi B Z_S \sqrt{\frac{2 \log \frac{4C}{\delta}}{n}}.
$$

The proof can be found in Appendix C.1 and the role played by $Z_S$ is investigated in Section 4.

### 3.3 SIMULTANEOUS LEARNING OF STATISTICAL PERFORMANCE AND FAIRNESS

We have established probabilistic bounds on both the statistical performance criterion and the fairness penalty. However, we wish to study both jointly, and determine the behaviour of the chosen fairness trade-off. As mentioned previously, this trade-off is very flexible and can accommodate multiple situations.

#### 3.3.1 BOUNDING LOSS AND FAIRNESS

First, we may wish to determine a probabilistic upper bound on $L_D^0(h) - L_S^0(h)$ and $\phi_D(h) - \phi_S(h)$ jointly. By a simple application of the union bound, we obtain the following result:

**Proposition 4.** *With probability at least $1 - \delta$, it holds* jointly *that*

$$
\sup_{h \in \mathcal{H}} \left| L_D^0(h) - L_S^0(h) \right| \leq 2R(\ell^0 \circ \mathcal{H} \circ \mathcal{S}) + B\sqrt{\frac{2 \log \frac{4}{\delta}}{n}}
$$

$$
\sup_{h \in \mathcal{H}} |\phi_D(h) - \phi_S(h)| \leq 2K^\phi \sum_{a=1}^{C} R(\ell^+ \circ \mathcal{H} \circ \mathcal{N}_a^+) + R(\ell^- \circ \mathcal{H} \circ \mathcal{N}_a^-)
$$

$$
+ K^\phi B Z_S \sqrt{\frac{2 \log \frac{8C}{\delta}}{n}}.
$$

#### 3.3.2 BOUNDING THE TRADE-OFF

As mentioned above, the existence of various impossibility results and the empirical findings showing that statistical performance tends to decrease as fairness requirements increase have highlighted the interest for partial debiasing methods. It is however important to be able to determine the generalisation properties that such objectives can lead to. We can perform a similar analysis on the trade-off objective $\mathcal{L}_D(h)$ by using the same arguments to get an upper bound on this objective.

**Proposition 5.** *With probability at least $1 - \delta$, it holds*

$$
\sup_{h \in \mathcal{H}} |\mathcal{L}_D(h) - \mathcal{L}_S(h)| \leq 2R(\ell^0 \circ \mathcal{H} \circ \mathcal{S}) + B\sqrt{\frac{2 \log \frac{4}{\delta}}{n}}
$$

$$
+ 2\lambda K^\phi \sum_{a=1}^{C} R(\ell^+ \circ \mathcal{H} \circ \mathcal{N}_a^+) + R(\ell^- \circ \mathcal{H} \circ \mathcal{N}_a^-)
$$

$$
+ \lambda K^\phi B Z_S \sqrt{\frac{2 \log \frac{8C}{\delta}}{n}}
$$

$$
\leq 2 \sum_{a=1}^{C} \left\{ \frac{n_a^+}{n} R(\ell^0 \circ \mathcal{H} \circ \mathcal{N}_a^+) + \frac{n_a^-}{n} R(\ell^0 \circ \mathcal{H} \circ \mathcal{N}_a^-) \right.
$$

$$
\left. + \lambda K^\phi R(\ell^+ \circ \mathcal{H} \circ \mathcal{N}_a^+) + \lambda K^\phi R(\ell^- \circ \mathcal{H} \circ \mathcal{N}_a^-) \right\}
$$

$$
+ B \left( 1 + \lambda K^\phi Z_S \right) \sqrt{\frac{2 \log(8C/\delta)}{n}}.
$$

The upper bound can be understood as the sum of individual contributions plus the usual $O\left(\sqrt{\log(1/\delta)/n}\right)$ factor. Let us now establish a related inequality, linking the distribution value of the trade-off under the distribution optimum and under the sample optimum. Let us denote by $h_T^* =_{h\in\mathcal{H}} \mathcal{L}_T(h)$ for $T \in \{D, S\}$ an optimal classifier derived on either the underlying distribution or the sample distribution of the objectives $\mathcal{L}_T$. This is an important issue for learnability of fairness as it provides some guarantees on how far the actual optimal trade-off is from the one we would obtain by picking the optimum computed on the sample $\mathcal{S}$. This result provides a fairness pendant to the usual case in statistical learning (see for instance Theorem 26.5 in Shalev-Shwartz & Ben-David (2014)).

**Theorem 1.** *With probability at least* $1 - \delta$, *it holds*

$$0 \leq \mathcal{L}_D(h_S^*) - \mathcal{L}_D(h_D^*) \leq UB_{\mathcal{S}} + B\left(1 + \lambda Z_S K^\phi\right) \frac{3}{\sqrt{2}}\sqrt{\frac{\log(16C/\delta)}{n}}, \qquad (8)$$

*where* $h_T^* =_{h\in\mathcal{H}} \mathcal{L}_T(h)$ *for* $T \in \{D, S\}$, *and* $UB_{\mathcal{S}} := 2\sum_{a=1}^{C}\left\{\frac{n_a^+}{n}R(\ell^0 \circ \mathcal{H} \circ \mathcal{N}_a^+) + \frac{n_a^-}{n}R(\ell^0 \circ \mathcal{H} \circ \mathcal{N}_a^-) + \lambda K^\phi R(\ell^+ \circ \mathcal{H} \circ \mathcal{N}_a^+) + \lambda K^\phi R(\ell^- \circ \mathcal{H} \circ \mathcal{N}_a^-)\right\}$.

The proof is given in Appendix A. What is particularly interesting about this result is the fact that the upper bound remains the same as in Proposition 5, and only the $O\left(\sqrt{\log(1/\delta)/n}\right)$ factor changes. Let us point out that this result is *practical* in nature in the sense that one would use $h_S^*$ in practice but would still look for out-of-sample generalisation, hence the importance of the term $\mathcal{L}_D(h_S^*)$.

## 4 (SOME) PRACTICAL CONSEQUENCES OF PAC FOR FAIRNESS TRADE-OFFS

### 4.1 DECOMPOSING UPPER BOUNDS ON GENERALISATION

One is led to decompose the overall loss in terms of each category's contributions to the overall learning upper bound.

**Definition 3.** The (half-)contribution –denoted by $\overline{R}_{\mathcal{N}_a}$– of each category $a = 1, \cdots, C$, to the fairness trade-off learning upper bound is

$$\begin{aligned}\overline{R}_{\mathcal{N}_a} \quad := \quad & \frac{n_a^+}{n}R(\ell^0 \circ \mathcal{H} \circ \mathcal{N}_a^+) + \frac{n_a^-}{n}R(\ell^0 \circ \mathcal{H} \circ \mathcal{N}_a^-) \\ & +\lambda K^\phi R(\ell^+ \circ \mathcal{H} \circ \mathcal{N}_a^+) + \lambda K^\phi R(\ell^- \circ \mathcal{H} \circ \mathcal{N}_a^-).\end{aligned}$$

*Remark* 2. This structure is quite interesting as it shows that the first two terms representing contributions to the statistical loss upper bound are weighted by their proportions of the overall sample. Thus, even if the Rademacher complexity of the category is high, it can be counterbalanced by the fact that it will not affect the overall average loss. On the other hand, the contributions coming from the fairness part are unweighted. In this case, a low sample size would usually lead to a higher Rademacher complexity.

Now, $\overline{R}_{\mathcal{N}_a}$ can be decomposed into finer components:

$$\overline{R}_{\mathcal{N}_a} = \overline{R}_{\mathcal{N}_a}^{0,+} + \overline{R}_{\mathcal{N}_a}^{0,-} + \overline{R}_{\mathcal{N}_a}^{+} + \overline{R}_{\mathcal{N}_a}^{-}, \qquad (9)$$

with obvious notations. One can then determine the contributions coming from the union of certain categories or of positives or negatives.

### 4.2 $Z_S$ AS SAMPLE CONCENTRATION

First, let us remark that the usefulness of probabilistic upper bounds comes from the fact that the ones that have been established only rely on observable quantities coming from the sample under consideration. This is powerful as a practical check, but it also suggests a way of amending the initial minimisation exercise by including the upper bound to the sample component, i.e., optimise $\mathcal{L}_D(h) + UB_{\mathcal{S}}(h)$.

Second, a quantity that is ubiquitous in our analysis (see, for instance, Theorem 1), is $Z_S = \sum_{a=1}^{C} \sqrt{\frac{n}{n_a^+}} + \sqrt{\frac{n}{n_a^-}}$. As noticed previously, this is a measure of concentration within the sample that only depends on in-sample data. The more unequal the count of categories' sample sizes, the higher $Z_S$. This can be formalised in the following proposition:

**Proposition 6.** *The constant $Z_S$ can be expressed in terms of the empirical class sample proportions:*

$$Z_S = \sum_{a=1}^{C} \left[ (\widehat{p_a^+})^{-\frac{1}{2}} + (\widehat{p_a^-})^{-\frac{1}{2}} \right] = \frac{2C}{\sqrt{M_{-1/2}(\widehat{\mathbf{p}})}}, \tag{10}$$

*where $M_\alpha(\widehat{\mathbf{p}})$ is the $\alpha$-generalised mean [1] of the vector $\widehat{\mathbf{p}} = (\widehat{p_a^\pm})_a^\pm$. In addition, under the assumption that $n_a^\pm \geq 1$ for all $a = 1, \cdots, C$, and $n > 2C - 1$, the following inequality holds:*

$$(2C)^{3/2} \leq Z_S \leq (2C - 1)\sqrt{n} + \sqrt{\frac{n}{n - (2C - 1)}}. \tag{11}$$

The proof is straightforward and left to the reader. The bounds on $Z_S$ offer two take-aways. First, the lower bound grows super-linearly as a function of the number of categories, which is problematic especially to tackle *intersectionality*. Second, the upper bound on $Z_S/\sqrt{(n)}$ converges to a *non-zero* constant as $n$ goes to infinity. This is an extreme scenario whereby a category can be made up of one individual only, but this stresses the fact that class imbalance can hinder learning in the context of fairness. $Z_S$ is model-agnostic and thus useful to determine *ex-ante* how much debiasing can be applied on a given dataset.

## 5 Asymptotic regime

One of the possible issues of using PAC learning in practice is the fact that it tends to consider worst-case scenarios (e.g., by deriving probabilistic inequalities over a whole space of possible classifiers). To mitigate this possible drawback, we adopt a very different viewpoint and set ourselves in a framework which generalises Proposition 1 in Agrawal et al. (2020). Our objective is to derive a limiting distribution for the rescaled fairness trade-off after having trained *one* particular classifier, which we then consider fixed. In doing so, we can thus assess the different contributions to the limiting variance *given* the choice of classifier. In addition to corresponding to real-life modelling, this also has the benefit of allowing for *closed-form* results (as opposed to upper bounds in PAC learning). Both approaches are –as will be shown– very complementary.

Let us say a few words about the idealised data generating process that hypothetically produces our observations. In particular, we consider that we have an infinite population of independently and identically distributed datapoints, according to a mixture distribution with probability weights $p_a^\pm$. Conditional on $a$ and $y = \pm 1$, we can then draw **x**. All draws are independent. Let us introduce some additional notation: the intra-category variances are given by $(\sigma_a^{0,\pm})^2 = \mathbb{V}_{s=a, y=\pm 1} \left[ \ell^0(h, z) \right]$, $(\sigma_a^\pm)^2 = \mathbb{V}_{s=a, y=\pm 1} \left[ \ell^\pm(h, z) \right]$. Furthermore, $K_{D,a}^\pm = \partial_{x_a^\pm} \phi_D(h)$, i.e., $K_{D,a}^\pm$ is the entry corresponding to $a^\pm$ in the gradient $\nabla \phi_D(h)$. We can now introduce our main result, which extends Agrawal et al. (2020) to a multi-category and multi-dimensional setting.

**Theorem 2.** *Let $h \in \mathcal{H}$, under mild assumptions, we have the following convergence in distribution*

$$\sqrt{n} \left( \mathcal{L}_S(h) - \mathcal{L}_D(h) \right) \xrightarrow[n \to +\infty]{d} N \left( 0, \mathbb{V}_{\lim}(h) \right), \tag{12}$$

*where*

$$\mathbb{V}_{\lim}[h] = \sum_{a^\pm} p_a^\pm \left( \sigma_a^{0,\pm} \right)^2 + \lambda^2 \frac{(K_a^\pm)^2}{p_a^\pm} \left( \sigma_a^\pm \right)^2 + \text{covariance terms} \tag{13}$$

The exact assumptions and the proof can be found in Appendix B. In a nutshell, the variance of the limiting trade-off distribution is the addition of

---

[1] The $\alpha$-generalised mean of $K$ numbers $x_1, \cdots, x_K$ is given by $\left( \frac{1}{K} \sum_{k=1}^{K} x_k^\alpha \right)^{1/\alpha}$.

- The sum of intra-category $\ell^{0,\pm}$ losses weighted by their true probabilities $p_a^\pm$;
- The sum of intra-category fairness $\ell^\pm$ losses weighted by their *inverse* probabilities $p_a^\pm$, as in the uniform convergence framework. In addition, the constant $(K_a^\pm)^2$ represents the sensitivity of the fairness loss function $\phi$ and is bounded from above by $[K^\phi]^2$ in these case where $\phi$ is $K^\phi$-Lipschitz continuous, while $\lambda$ is the hyper-parameter governing the trade-off between statistical and fairness performances.

Qualitatively, the results obtained in the asymptotic and uniform frameworks are similar and point to the conclusion that to learn fairness, one must start by learning category samples. A low probability $p_a^\pm$ may not impact the variance of the overall statistical loss but it may lead to a very poor understanding of fairness trade-offs.

## 6   EXPERIMENTAL RESULTS

In this Section, we investigate on real datasets the validity of the theoretical results that we have derived. Here, we solve (via L-BFGS) for a logistic regression with an additional constraint on fairness. Concretely, the objective function is defined as:

$$\min_{\boldsymbol{\beta} \in \mathbb{R}^d} \frac{1}{n} \sum_{i=1}^{n} \ell(y_i, \mathbf{x}_i; \boldsymbol{\beta}) + \lambda \left[ \frac{1}{n_0} \sum_{j=1}^{n_0} \ell(y_j, \mathbf{x}_j; \boldsymbol{\beta}) - \frac{1}{n_1} \sum_{j'=1}^{n_1} \ell(y_{j'}, \mathbf{x}_{j'}; \boldsymbol{\beta}) \right]^2, \qquad (14)$$

where $\ell$ is the logistic loss function and the groups 0 and 1 correspond to two different classes of a protected attribute. We have considered three datasets with binary protected attributes and binary outcomes, shown in Table 2, and applied a standard 80–20 train-test split.

| Dataset name | Reference | Protected attribute | Binary outcome |
|---|---|---|---|
| Adult | Dua & Graff (2017) | sex | income exceeds 50$K/year |
| Loan defaults | I-Cheng Yeh (2009) | sex | default payment next month |
| German credit | Dua & Graff (2017)[2] | sex | creditability |

Table 2: Data sets description

Figures 1 and 2 illustrate the trade off between average loss and disparity for a range of $\lambda$ values. [3]

On the *train test* (Figure 1, there is a clear dependence between the value of $\lambda$ and the respective average loss and disparity between groups. As expected, an increase in $\lambda$ tends to decrease disparity, while increasing the average loss, and vice versa. However, this is not always the case on the *test set*, as some intermediate values of $\lambda$ achieve both lower average loss and lower disparity, whereas more (or totally) debiased models clearly overfit.

### 6.1   IN-SAMPLE RESULTS

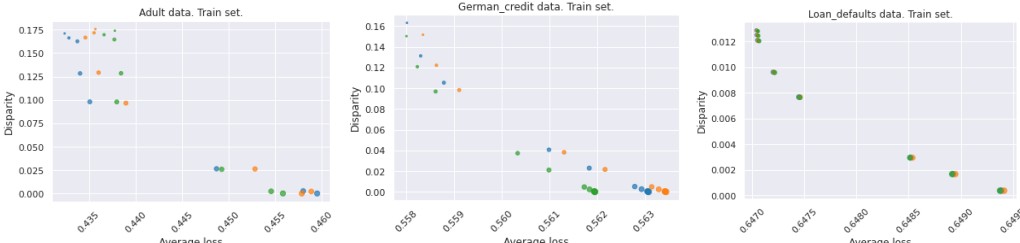

Figure 1: Trade off between disparity and average loss on adult, german credit, and loan defaults data sets. Each colour corresponds to a different initialisation, while the symbol's size corresponds to the weight $\lambda$ on disparity in the overall loss function. Results are measures on the *train set*.

---

[3]Specifically $\{0.01, 0.05, 0.1, 0.5, 1, 5, 10, 50, 100, 500\}$. Each distinct color represents a different initialization, while the size of the point is a monotonically increasing function of the respective $\lambda$.

## 6.2 OUT-OF-SAMPLE RESULTS

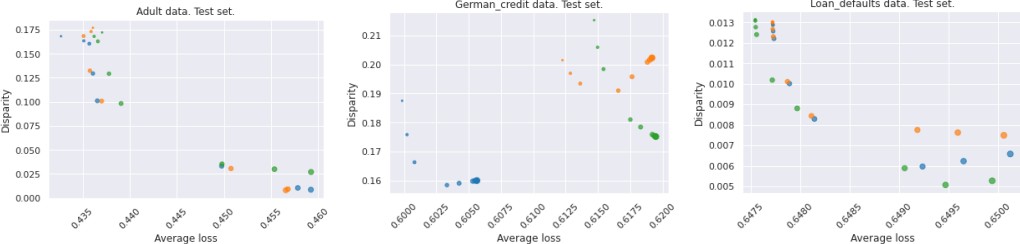

Figure 2: Trade-off between disparity and average loss on Adult, German credit, and Loan defaults datasets. Each colour corresponds to a different initialisation, while the symbol's size corresponds to the weight $\lambda$ on disparity in the overall loss function. Results are measured on the *test set*.

## 6.3 STANDARD DEVIATION SCALING

We experiment with class imbalance to see how the solution to the original problem changes as imbalance grows, in line with Theorem 2. To do so, we downsample observations from one class, while keeping the original number of observations in the other class. As predicted, on the test set, the loss variance increases with class imbalance, as shown in Figure 3[4].

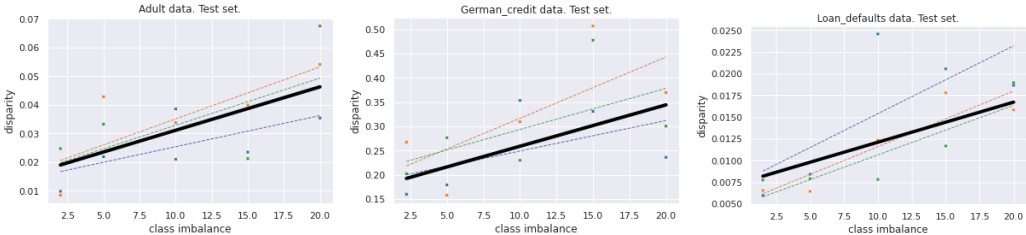

Figure 3: Fairness uncertainty versus class imbalance ratio in protected characteristic on Adult, German credit, and Loan defaults datasets. Results are measured on the test set.

## 7 CONCLUSION

The analysis that we have conducted from two different angles, namely learning with uniform convergence and asymptotics, leads to the same overall qualitative assessment. While we have derived probabilistic upper bounds to prove learning guarantees on the one hand and convergence to a limiting distribution on the other, a striking feature of learning fairness trade-offs is the fundamental difference of regimes between the usual statistical performance criterion measured on the whole sample and the fairness penalty that examines relationships between sub-samples.

Indeed, fairness is not about learning an average distribution, quite the contrary, in the sense that it requires a fine understanding of differences across conditional distributions. Intuitively, in the usual case of an overall statistical loss function, if a category only represents a small portion of the sample, then it also only constitutes a small fraction of the overall loss. On the opposite, from a fairness viewpoint, this implies that it is difficult to make any type of statements regarding the relationship between this particular category and other categories, hence a high risk for a given fairness objective. We have also shown with an empirical investigation that our results translate to practical cases and that small sample sizes or class imbalance can lead to spurious empirical results.

---

[4]Here, points of the same color correspond to the same random seed. Different random seeds are needed to obtain several variants of the downsampled data. The dotted color lines are the plain linear least square estimate through the points of the same color, while the bold black line is least square estimate using data points from all seeds. $\lambda$ is fixed at 50.

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

In the appendix, we indicate the proofs of the two theorems in the main text, Thm 1 and Thm 2.

## A  SOME APPLICATIONS OF FAIR LEARNING

In this Section, we apply our results to two particular loss functions, namely the Calders-Verwer gap and the Bayes gap.

### A.1  EXAMPLE 1: LEARNING DISPARITY

Of particular interest when measuring fairness is the discrepancy that can be observed amongst different groups, for example in terms of a loss $L$ that captures some properties we wish to ascertain. This could be the difference in false negative rates between categories $a$ and $a'$, where $a$ is usually chosen to be a reference group. This concept holds for both the true distribution and the sample distribution.

**Definition 4.** The (demographic) disparity, or Calders-Verwer gap Calders & Verwer (2010); Chen et al. (2019), between groups $a$ and $a'$ with respect to loss $L$ can be defined as

$$\Delta_{T,a,a'}^L(h) = L_{T,a}(h) - L_{T,a'}(h), \tag{15}$$

for $T \in \{D, S\}$.

Note that it is also quite usual to consider the absolute value of disparity, $\left|\Delta_{T,a,a'}^L(h)\right|$ indicating whether there is equality or not (and magnitude thereof), rather than giving a sign (which relies more on how the baseline category has been chosen). Indeed, trivially, disparity is asymmetric whereas the absolute disparity is symmetric.

It is therefore important to be able to establish the learnability of this widely used metric, which we do in the following proposition.

**Proposition 7.** *With probability at least $1 - \delta$, it holds*

$$\sup_{h \in \mathcal{H}} \left| \Delta_{D,a,a'}^L(h) - \Delta_{S,a,a'}^L(h) \right| \leq 2\left\{ R(\ell \circ \mathcal{H} \circ \mathcal{N}_a) + R(\ell \circ \mathcal{H} \circ \mathcal{N}_{a'}) \right\}$$

$$+ B \sqrt{\frac{2\log\left(\frac{4}{\delta}\right)}{n_a + n_{a'}}} \left[ \sqrt{\frac{n_a + n_{a'}}{n_a}} + \sqrt{\frac{n_a + n_{a'}}{n_{a'}}} \right].$$

*Proof.* We start by observing that by sub-additivity of the sup, we have

$$\sup_{h \in \mathcal{H}} \left| \Delta_{D,a,a'}^L(h) - \Delta_{S,a,a'}^L(h) \right| \leq \sup_{h \in \mathcal{H}} |L_{D,a}(h) - L_{S,a}(h)| + \sup_{h \in \mathcal{H}} |L_{D,a'}(h) - L_{S,a'}(h)|.$$

We can then conlude thanks to Proposition 1 and the union bound. □

What this upper bound suggests is that in order to learn about disparity, one has to learn the two categories $a$ and $a'$. If one category has either a large Rademacher complexity or a low count, the upper bound will increase. This result enables us to derive bounds on quantities of interest such as $\left| \Delta_{D,a,a'}^L(h_S^*) - \Delta_{S,a,a'}^L(h_S^*) \right|$ that follows directly from Proposition 7.

## A.2 Example 2: Learning bias

In this Section, we understand bias in a very specific (and non-fairness related) way, as in Chen et al. (2018). In short, one can consider bias –in this context– as the difference between the loss incurred by category $a$ when the classifier is determined by minimising the loss function over the entire sample and the loss in category $a$ when the optimal classifier is derived on a standalone basis (i.e., the classifier explicitly takes into the account the attribute $a$).

If it is possible to use the characteristics $s$ directly in the model, then one can calibrate an optimal classifier on each category:

$$h_D^*(\mathbf{x}, s) = \sum_{a=1}^{C} \mathbf{1}_{\{s=a\}} h_{D,a}^*(\mathbf{x}). \tag{16}$$

However, we usually do not have access to the characteristic $s$ or cannot include it as a feature (for instance to avoid disparate treatment), so that –in general– $h_D(\mathbf{x}, s) = h_D^*(\mathbf{x})$ for all $s = 1, \cdots, C$. Consequently, we can define the Bayes gap as the difference between the loss obtained between the

**Definition 5.** The *Bayes gap* for category $a$ is given by

$$\Gamma_{T,a} = L_{T,a}(h_S^*) - L_{T,a}(h_{S,a}^*), \tag{17}$$

for $T \in \{D, S\}$.

In particular, $\Gamma_{S,a} \geq 0$. $\Gamma_{\cdot,a}$ represents the added loss incurred by category $a$ due to the consideration of other categories while selecting the classifier $h$.

**Proposition 8.** *The true Bayes gap, $\Gamma_{D,a} = L_{D,a}(h_S^*) - L_{D,a}(h_{S,a}^*)$ can be learnt as*

$$|\Gamma_{D,a} - \Gamma_{S,a}| \leq 4R(\ell \circ \mathcal{H} \circ \mathcal{N}_a) + 2B\sqrt{\frac{2\log(2/\delta)}{2n_a}}, \tag{18}$$

*with probability at least $1 - \delta$.*

*Proof.* The proof is fairly straightforward and simply decomposes the Bayes gap in terms of easier building blocks:

$$L_{D,a}(h_S^*) - L_{D,a}(h_{S,a}^*) = L_{D,a}(h_S^*) - L_{S,a}(h_S^*) + L_{S,a}(h_S^*) - L_{S,a}(h_{S,a}^*)$$
$$+ L_{S,a}(h_{S,a}^*) - L_{D,a}(h_{S,a}^*),$$

leading to

$$\Gamma_{D,a} - \Gamma_{S,a} = L_{D,a}(h_S^*) - L_{S,a}(h_S^*) + L_{S,a}(h_{S,a}^*) - L_{D,a}(h_{S,a}^*).$$

We thus obtain that

$$
\begin{aligned}
|\Gamma_{D,a} - \Gamma_{S,a}| &\leq 2 \sup_{h \in \mathcal{H}} |L_{D,a}(h) - L_{S,a}(h)| \\
&\leq 4R(\ell \circ \mathcal{H} \circ \mathcal{N}_a) + 2B\sqrt{\frac{2\log(2/\delta)}{2n_a}},
\end{aligned}
$$

where the second inequality holds with probability at least $1 - \delta$ by Proposition 1. $\qquad\square$

## B  OUT-OF-SAMPLE VARIANCE SCALING

In this Section, we consider a surrogate example to illustrate some real-world consequences of Theorem 2. The behaviour of a number of debiasing methods, both in- and out-of-sample, has been studied on real and surrogate data in the literature (Zafar et al. (2017); Agrawal et al. (2020); Oneto et al. (2020) for instance), we thus only provide an illustration of our result on a simple test case, to highlight the role played by sampling variance in dealing with fairness metrics. In keeping with the previous section, we suppose that we have chosen a given classifier $\widehat{y} = h(\mathbf{x}, s)$ and are now considering its statistical properties on a (large) out-of-sample dataset.

### B.1  PREDICTIVE PARITY AND ITS LIMITING VARIANCE

In our simplified experimental setup, we consider $C = 2$, and set $\ell^0 = 0$ with $\lambda = 1$, i.e., we only look at a fairness criterion, and choose a ratio $\tau_{\text{PP}}$ linked to predictive parity (see Table 1):

$$\phi_D(h) = \tau_{D,\text{PP}} := \frac{\mathbb{P}(\widehat{y} = 1 | y = 1, s = 1)}{\mathbb{P}(\widehat{y} = 1 | y = 1, s = 2)}. \tag{19}$$

The perfectly fair case corresponds to $\tau_{D,\text{PP}} = 1$. We also define the corresponding sample version of this criterion, using the samples $\mathcal{N}_a^+$, $a = 1, 2$:

$$\phi_S(h) = \tau_{S,\text{PP}} = \frac{\frac{1}{n_1^+} \sum_{i \in \mathcal{N}_1^+} \mathbf{1}_{\{\widehat{y}_i = 1\}}}{\frac{1}{n_2^+} \sum_{i \in \mathcal{N}_2^+} \mathbf{1}_{\{\widehat{y}_i = 1\}}}. \tag{20}$$

In this case, one can compute $\mathbb{V}_{\lim}(h)$ directly as

$$\mathbb{V}_{\lim}(h) = \frac{1 - \mathbb{P}(\widehat{y} = 1 | y = 1, s = 1)}{\mathbb{P}(\widehat{y} = 1 | y = 1, s = 1)} \frac{1}{\pi_1} + \frac{1 - \mathbb{P}(\widehat{y} = 1 | y = 1, s = 2)}{\mathbb{P}(\widehat{y} = 1 | y = 1, s = 2)} \frac{1}{\pi_2}, \tag{21}$$

where $\pi_1 = \frac{p_1^+}{p_1^+ + p_2^+}$ and $\pi_2 = 1 - \pi_1 = \frac{p_2^+}{p_1^+ + p_2^+}$.

### B.2  SIMULATION SET-UP AND RESULTS

In our simulations, we use the fixed values $\mathbb{P}(\widehat{y} = 1 | y = 1, s = 1) = 0.95$ and $\mathbb{P}(\widehat{y} = 1 | y = 1, s = 2) = 0.99$, leading to $\tau_{D,\text{PP}} = 0.96$. We vary the parameters $m = n_1^+ + n_2^+$, which represents the overall sample size available to us to compute $\tau_{S,\text{PP}}$, and $\pi_1$, which is the effective proportion of class 1 versus class 2. Throughout this exercise, we simulate $N = 1000$ times the experiment. Our baseline case is $m = 1,000$ and $\pi_1 = 10\%$.

From Figure 4 (a), one recovers the usual behaviour due to sample size: as $m$ increases, the empirical distribution of $\tau_{S,\text{PP}}$ becomes more peaked. A small sample size (such as $m = 100$) would yield poor accuracy and one can observe that the density is actually bi-modal.

The behaviour with respect to sample size can also be checked by comparing the empirical distribution of $\frac{\tau_{S,\text{PP}}^{(k)} - \tau_{D,\text{PP}}}{\sqrt{\mathbb{V}_{\lim}(h)}}$, where $k = 1, \cdots, N$ is the experiment run, with a standard Gaussian distribution, as predicted by Proposition 2. It can be seen in Figure 5 that there is a good adequacy between both distributions.

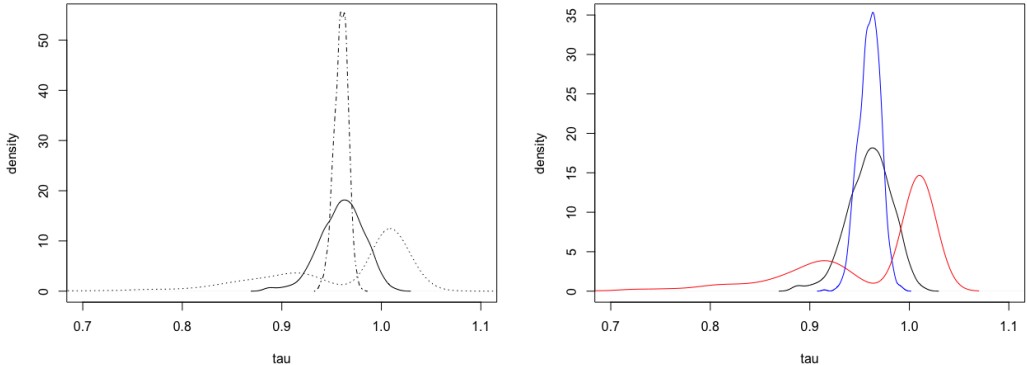

Figure 4: **(a)** Empirical densities of $\tau_{S,\text{PP}}$ for $m = 100$ (*dotted*), 1000 (*black*) and 10000 (*dashed*) at $\pi_1 = 10\%$. **(b)** Empirical densities of $\tau_{S,\text{PP}}$ for $\pi_1 = 1\%$ (*red*), 10% (*black*) and 50% (*blue*) at $m = 1000$.

The most salient feature, however, is the behaviour of $\tau_{S,\text{PP}}$'s distribution with respect to category probability, as described in Figure 4 (b), where one can see that as the sample becomes balanced, the variance of the estimator $\tau_{S,\text{PP}}$ decreases. When $\pi_1$ is very small, say 1%, one recovers the bi-modal distribution that comes with a small sample.

### B.3   KEY OBSERVATION

In a nutshell, low sample sizes and class imbalance can lead to poor generalisation. Indeed, by considering the cases $m = 100, \pi_1 = 10\%$ and $m = 1000, \pi_1 = 1\%$, it is clear that one cannot draw any conclusion regarding the presence of bias nor in terms of which group is the advantaged group. Bi-modality is a particularly interesting characteristic of the empirical distribution: there is a mode strictly below 1 and a mode strictly above 1.

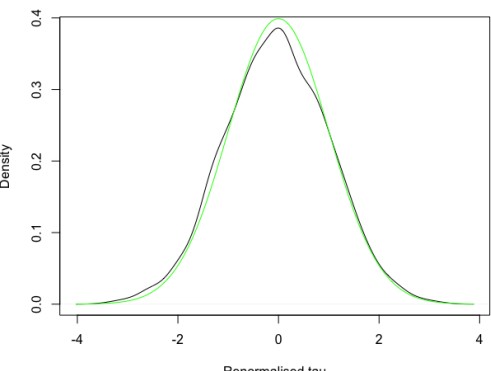

Figure 5: Normalised density of $\tau_{D,\text{PP}}$ (in *black*) and theoretical limiting distribution (in *green*) for $m = 1,000$ and $\pi_1 = 10\%$.

# C PROOFS

## C.1 PROOF OF PROPOSITION 3

**Proposition 3** *Under the assumption that $\phi$ is $K^\phi$-Lipschitz, it holds, with probability at least $1 - \delta$, that*

$$\sup_{h \in \mathcal{H}} |\phi_D(h) - \phi_S(h)| \leq 2K^\phi \sum_{a=1}^{C} R(\ell^+ \circ \mathcal{H} \circ \mathcal{N}_a^+) + R(\ell^- \circ \mathcal{H} \circ \mathcal{N}_a^-)$$

$$+ K^\phi B Z_S \sqrt{\frac{2 \log \frac{4C}{\delta}}{n}}.$$

*with $Z_S := \sum_{a=1}^{C} \sqrt{\frac{n}{n_a^+}} + \sqrt{\frac{n}{n_a^-}}$.*

*Proof.* Using the hypothesis that $\phi$ is $K^\phi$-Lipschitz, it comes

$$|\phi_D(h) - \phi_S(h)| \leq K^\phi \sum_{a=1}^{C} \left| L_{D,a}^+(h) - L_{S,a}^+(h) \right| + \left| L_{D,a}^-(h) - L_{S,a}^-(h) \right|$$

For any $\delta' \in (0,1)$ and any $a = 1, \cdots, C$ it holds

$$\sup_{h \in \mathcal{H}} \left| L_{D,a}^\pm(h) - L_{S,a}^\pm(h) \right| \leq 2R(\ell^\pm \circ \mathcal{H} \circ \mathcal{N}_a^\pm) + B \sqrt{\frac{2 \log \frac{2}{\delta'}}{n_a^\pm}},$$

whereby

$$\sup_{h \in \mathcal{H}} |\phi_D(h) - \phi_S(h)| \leq 2K^\phi \sum_{a=1}^{C} R(\ell^+ \circ \mathcal{H} \circ \mathcal{N}_a^+) + R(\ell^- \circ \mathcal{H} \circ \mathcal{N}_a^-)$$

$$+ K^\phi B \sum_{a=1}^{C} \sqrt{\frac{2 \log \frac{2}{\delta'}}{n_a^+}} + \sqrt{\frac{2 \log \frac{2}{\delta'}}{n_a^-}},$$

with probability at least $1 - 2C\delta'$, thanks to the union bound. To have $1 - \delta = 1 - 2C\delta'$, one can simply pick $\delta' = \delta/(2C)$. Now, we can write the last term as

$$\sum_{a=1}^{C} \sqrt{\frac{2 \log \frac{2}{\delta'}}{n_a^+}} + \sqrt{\frac{2 \log \frac{2}{\delta'}}{n_a^-}} = \sqrt{\frac{2 \log \frac{2}{\delta'}}{n}} \sum_{a=1}^{C} \sqrt{\frac{n}{n_a^+}} + \sqrt{\frac{n}{n_a^-}}$$

$$= Z_S \sqrt{\frac{2 \log \frac{2}{\delta'}}{n}},$$

hence the final result. $\square$

## C.2 PROOF OF THEOREM 1

**Theorem 1** *With probability at least $1 - \delta$, it holds*

$$0 \leq \mathcal{L}_D(h_S^*) - \mathcal{L}_D(h_D^*) \leq UB_S + B \left( 1 + \lambda Z_S K^\phi \right) \frac{3}{\sqrt{2}} \sqrt{\frac{\log(16C/\delta)}{n}}, \tag{22}$$

*Proof.* For the sake of clarity, we divide the proof into different steps.

**Step 1.** We can start by rewriting

$$\mathcal{L}_D(h_S^*) - \mathcal{L}_D(h_D^*) = \mathcal{L}_D(h_S^*) - \mathcal{L}_S(h_S^*) + \mathcal{L}_S(h_S^*) - \mathcal{L}_S(h_D^*) + \mathcal{L}_S(h_D^*) - \mathcal{L}_D(h_D^*)$$

$$\leq \mathcal{L}_D(h_S^*) - \mathcal{L}_S(h_S^*) + \mathcal{L}_S(h_D^*) - \mathcal{L}_D(h_D^*),$$

since, by definition, $\mathcal{L}_S(h_S^*) - \mathcal{L}_S(h_D^*) \le 0$.

**Step 2.** Now, $\mathcal{L}_D(h_S^*) - \mathcal{L}_S(h_S^*) \le \sup_{h \in \mathcal{H}} |\mathcal{L}_D(h) - \mathcal{L}_S(h)|$ on the one hand, and, on the other hand, we have

$$
\begin{aligned}
|\mathcal{L}_S(h_D^*) - \mathcal{L}_D(h_D^*)| \le{}& \left| L_S^0(h_D^*) - L_D^0(h_D^*) \right| \\
&+ \lambda K^\phi \sum_{a=1}^C \left\{ \left| L_{D,a}^+(h_D^*) - L_{S,a}^+(h_D^*) \right| + \left| L_{D,a}^-(h_D^*) - L_{S,a}^-(h_D^*) \right| \right\}.
\end{aligned}
$$

**Step 3.** The key insight is to notice that the right-hand side only involves $h_D^*$, which does *not* depend on the sample $\mathcal{S}$, hence one can apply the Hoeffding inequality directly. Thus, by Hoeffding's inequality, for each $a$, with probability at least $1 - \delta''$,

$$
\left| L_{D,a}^\pm(h_D^*) - L_{S,a}^\pm(h_D^*) \right| \le B \sqrt{\frac{\log(2/\delta'')}{2n_a^\pm}}.
$$

Similarly, with probability at least $1 - \delta''$,

$$
\left| L_S^0(h_D^*) - L_D^0(h_D^*) \right| \le B \sqrt{\frac{\log(2/\delta'')}{2n}}.
$$

We can then infer that with probability at least $1 - (2C + 1)\delta''$

$$
\begin{aligned}
|\mathcal{L}_S(h_D^*) - \mathcal{L}_D(h_D^*)| \le{}& B \sqrt{\frac{\log(2/\delta')}{2n}} \\
&+ \lambda B K^\phi \sum_{a=1}^C \sqrt{\frac{\log(2/\delta'')}{2n_a^+}} + \sqrt{\frac{\log(2/\delta'')}{2n_a^-}} \\
\le{}& B \left(1 + \lambda Z_S K^\phi\right) \sqrt{\frac{\log(2/\delta'')}{2n}}.
\end{aligned}
$$

**Step 4.** From the previous proposition, it comes that with probability at least $1 - \delta'$

$$
\sup_{h \in \mathcal{H}} |\mathcal{L}_D(h) - \mathcal{L}_S(h)| \le \mathrm{UB}_\mathcal{S} + B \left(1 + \lambda Z_S K^\phi\right) \sqrt{\frac{2 \log(8C/\delta')}{n}}.
$$

Finally, by choosing $\delta' = \delta/2$ and $\delta'' = \delta/(2(2C + 1))$,

$$
\begin{aligned}
\mathcal{L}_D(h_S^*) - \mathcal{L}_D(h_D^*) \le{}& \mathrm{UB}_\mathcal{S} + B \left(1 + \lambda Z_S K^\phi\right) \left[ \sqrt{\frac{2 \log(16C/\delta)}{n}} + \sqrt{\frac{\log(4(2C + 1)/\delta)}{2n}} \right] \\
\le{}& \mathrm{UB}_\mathcal{S} + B \left(1 + \lambda Z_S K^\phi\right) \frac{3}{\sqrt{2}} \sqrt{\frac{\log(16C/\delta)}{n}},
\end{aligned}
$$

with probability at least $1 - \delta$. $\qquad\square$

### C.3 PROOF OF THEOREM 2

**Theorem 2** *Let $h \in \mathcal{H}$, under the assumption that $p_a^\pm > 0$ for all $a$, and that $\nabla \phi_D(h)$ exists and is non-zero, we have the following convergence in distribution*

$$
\sqrt{n} \left( \mathcal{L}_S(h) - \mathcal{L}_D(h) \right) \xrightarrow[n \to +\infty]{d} N\left(0, \mathbb{V}_{\lim}(h)\right), \tag{23}
$$

*where*

$$
\begin{aligned}
\mathbb{V}_{\lim}(h) \;=\; & \sum_{a=1}^{C} p_a^+ \left(\sigma_a^{0,+}\right)^2 + \lambda^2 \frac{(K_a^+)^2}{p_a^+} \left(\sigma_a^+\right)^2 + 2\lambda K_a^+ cov_a^+ \left(\ell^0(h,z), \ell^+(h,z)\right) \\
& + \sum_{a=1}^{C} p_a^- \left(\sigma_a^{0,-}\right)^2 + \lambda^2 \frac{(K_a^-)^2}{p_a^-} \left(\sigma_a^-\right)^2 + 2\lambda K_a^- cov_a^- \left(\ell^0(h,z), \ell^-(h,z)\right) \\
& + \sum_{a=1}^{C} p_a^+ (1-p_a^+) \left(L_{D,a}^{0,+}\right)^2 + p_a^- (1-p_a^-) \left(L_{D,a}^{0,-}\right)^2 - 2 p_a^+ p_a^- L_{D,a}^{0,+} L_{D,a}^{0,-} \\
& - \sum_{a \neq a'} \Bigg( p_a^+ p_{a'}^+ L_{D,a}^{0,+} L_{D,a'}^{0,+} + p_a^- p_{a'}^- L_{D,a}^{0,-} L_{D,a'}^{0,-} \\
& \qquad\qquad + p_a^+ p_{a'}^- L_{D,a}^{0,+} L_{D,a'}^{0,-} + p_a^- p_{a'}^+ L_{D,a}^{0,-} L_{D,a'}^{0,+} \Bigg)
\end{aligned}
$$

*Proof.* For the sake of clarity, we divide this simple (but slightly tedious) proof into multiple steps. In this proof, we make the assumption that $\phi$ is differentiable with continuous gradient; this is not necessary but simplifies the exposition. The reader is referred to DasGupta (2008) for details regarding the delta method, the central limit theorem and different types of convergence.

**Step 1.** We start by rewriting the difference between both true and empirical trade-offs $\mathcal{L}_S$ and $\mathcal{L}_D$ for a given $h \in \mathcal{H}$. Thanks to the Taylor formula, there exists $\xi \in [0,1]$ such that

$$
\begin{aligned}
\phi_S(h) - \phi_D(h) \;=\; & \phi(\mathbf{L}_S^{\pm}(h)) - \phi(\mathbf{L}_D^{\pm}(h)) \\
=\; & \nabla\phi(\xi \mathbf{L}_D^{\pm}(h) + (1-\xi)\mathbf{L}_S^{\pm}(h))^T \left(\mathbf{L}_S^{\pm}(h) - \mathbf{L}_D^{\pm}(h)\right)
\end{aligned}
$$

We denote $K_{S,a}^{\pm}$ the partial differential of $\phi$ with respect to $x_a^{\pm}$, i.e., $K_{S,a}^{\pm} = \partial_{x_a^{\pm}} \phi(\xi \mathbf{L}_D^{\pm}(h) + (1-\xi)\mathbf{L}_S^{\pm}(h))^T \left(\mathbf{L}_S^{\pm}(h) - \mathbf{L}_D^{\pm}(h)\right)$. Some algebra then yields

$$
\begin{aligned}
& \sqrt{n}\left(\mathcal{L}_S(h) - \mathcal{L}_D(h)\right) \\
& = \sum_{a=1}^{C} \sqrt{\frac{n}{n_a^+}} \sqrt{n_a^+} \left(p_a^+ \left[L_{S,a}^{0,+}(h) - L_{D,a}^{0,+}(h)\right] + \lambda K_{S,a}^+ \left[L_{S,a}^+(h) - L_{D,a}^+(h)\right]\right) \\
& + \sum_{a=1}^{C} \sqrt{\frac{n}{n_a^-}} \sqrt{n_a^-} \left(p_a^- \left[L_{S,a}^{0,-}(h) - L_{D,a}^{0,-}(h)\right] + \lambda K_{S,a}^- \left[L_{S,a}^-(h) - L_{D,a}^-(h)\right]\right) \\
& \qquad\qquad + \sum_{a=1}^{C} \sqrt{n}\left(\frac{n_a^+}{n} - p_a^+\right) L_{S,a}^{0,+}(h) + \sum_{a=1}^{C} \sqrt{n}\left(\frac{n_a^-}{n} - p_a^-\right) L_{S,a}^{0,-}(h).
\end{aligned}
$$

**Step 2.** Since $\lim_{n \to +\infty} \frac{n_a^{\pm}}{n} = p_a^{\pm} > 0$ almost surely, this implies trivially that $\lim_{n \to +\infty} n_a^{\pm} = +\infty$ almost surely, for any $a$. We now recall that, thanks to the continuous mapping theorem, it holds that for any $n_a^{\pm}$, $\lim_{n \to +\infty} \sqrt{n/n_a^{\pm}} = 1/\sqrt{p_a^{\pm}}$ almost surely. Finally, $\lim_{n \to +\infty} K_{S,a}^{\pm} = K_{D,a}^{\pm}$ almost surely, since $\nabla\phi$ is continuous and $\lim_{n \to +\infty} \mathbf{L}_S^{\pm}(h) = \mathbf{L}_D^{\pm}(h)$ almost surely.

**Step 3.** Let us now consider each term $a^{\pm}$ separately.

$$
\begin{aligned}
& \sqrt{n_a^{\pm}} \left(p_a^{\pm} \left[L_{S,a}^{0,\pm}(h) - L_{D,a}^{0,\pm}(h)\right] + \lambda K_{S,a}^{\pm} \left[L_{S,a}^{\pm}(h) - L_{D,a}^{\pm}(h)\right]\right) \\
& = \sqrt{n_a^{\pm}} \left(\left[p_a^{\pm} L_{S,a}^{0,\pm}(h) + \lambda K_{D,a}^{\pm} L_{S,a}^{\pm}(h)\right] - \left[p_a^{\pm} L_{D,a}^{0,\pm}(h) + \lambda K_{D,a}^{\pm} L_{D,a}^{\pm}(h)\right]\right) \\
& \qquad\qquad + \lambda\left(K_{S,a}^{\pm} - K_{D,a}^{\pm}\right) \sqrt{n_a^{\pm}} \left[L_{S,a}^{\pm}(h) - L_{D,a}^{\pm}(h)\right].
\end{aligned}
$$

By the Central Limit Theorem (CLT), $\sqrt{n_a^{\pm}} \left[L_{S,a}^{\pm}(h) - L_{D,a}^{\pm}(h)\right] \xrightarrow[n \to +\infty]{d} N(0, [\sigma_a^{\pm}]^2)$, but since $\left(K_{S,a}^{\pm} - K_{D,a}^{\pm}\right) \xrightarrow[n \to +\infty]{a.s.} 0$, the whole term $\left(K_{S,a}^{\pm} - K_{D,a}^{\pm}\right) \sqrt{n_a^{\pm}} \left[L_{S,a}^{\pm}(h) - L_{D,a}^{\pm}(h)\right]$ goes to 0 almost surely.

All that remains is to apply the CLT to

$$\sqrt{n_a^{\pm}} \left( \left[ p_a^{\pm} L_{S,a}^{0,\pm}(h) + \lambda K_{D,a}^{\pm} L_{S,a}^{\pm}(h) \right] - \left[ p_a^{\pm} L_{D,a}^{0,\pm}(h) + \lambda K_{D,a}^{\pm} L_{D,a}^{\pm}(h) \right] \right),$$

and notice that $\mathbb{V}_{s=a,y=\pm 1} \left[ p_a^{\pm} \ell^{0,\pm}(h,z) + \lambda K_{D,a}^{\pm} \ell^{\pm}(h,z) \right] = [p_a^{\pm}]^2 [\sigma_a^{0,\pm}]^2 + \lambda^2 [K_{D,a}^{\pm}]^2 [\sigma_a^{\pm}]^2 + 2 p_a^{\pm} \lambda K_{D,a}^{\pm} \mathrm{cov}_a^{\pm} \left( \ell^0(h,z), \ell^{\pm}(h,z) \right)$.

Note that since each data point in the sample is independent, the CLT can be applied term by term (i.e., all $a^{\pm}$'s have independent limiting distributions).

**Step 4.** To finish the proof, it simply remains to consider the adjustments due to observing empirical proportions rather than the true class probabilities. We thus need to consider the limiting distribution of the vector $\sqrt{n} \left( \frac{n_a^{\pm}}{n} - p_a^{\pm} \right)_{\pm, a=1,\cdots,C}$. But this is simply the limiting distribution of empirical proportions in repeated trials of a multinoulli distribution, hence the result. $\square$

Let us point out that the non-vanishing assumption on the gradient of the fairness loss function $\phi$ can be relaxed, but leads to the need for a higher-order delta method (see DasGupta (2008)).

