# OpenReview forum: "On Learning with Fairness Trade-Offs"
_ICLR.cc/2022/Conference — ICLR 2022 Submitted_

### Official Review · Reviewer_9tuc · 2021-11-02

**Correctness:** 3
**Technical Novelty And Significance:** 2
**Empirical Novelty And Significance:** 2
**Recommendation:** 3
**Confidence:** 4

**Main Review:**

Strengths:
  - These bounds, although perhaps not surprising, are useful theoretical upper bounds for generalization under fairness.

Weakness:
  - The proof ideas are not very novel. Majority of the proofs are application of Rademacher inequalities and union bounds.
  - The "asymptotic regime" is quite limited in its discussion and analysis.
  - Theorem 2's improvement of Agrawal et al. (2020)'s result does not seem significant. The only change I can see is accounting for extra dimensions is derivatives and added summations. (Note: The proof of the citation is accessible through the arxiv tar.gz download). A side note: the reference to Proposition 1 does not match the latest arxiv release of Agrawal et al. (2020).
  - Proof of Proposition 6 would be appreciated in the appendix (particularly the upper bound). Also it seems you can trivially get a tighter upper bound of $ 2C\sqrt{n} $ (setting denominators to 1).
  - The experimentation's stated claim is to investigate the validity of the theoretical claims, however, 6.1 & 6.2 appear to only test the trade-off in regularization. A better direction would be perhaps to test out-of-distribution generalization bounds experimentally.
  - The class imbalance of 6.3 is never explained; and factor of just having less data in the experiment is not explored.
  - There are many notation choice, typos, and missing definitions which have made navigating the paper difficult. One especially prohibitive choice is the use of the $ \pm $ subscript, where it is unclear in many places if this is used to denote a specific equation; or is being used to define a $ + $ and $ - $ variant of the equation.

Other:
  - Just a note that the losses with subscript $ S $ is sometimes used as notation for subgroup loss in other work.
  - Definition 2 should clarify $ \sigma_{i} $.
  - Type facing of Rademacher complexity verses constants $ R $ should be clarified.
  - Missing $ \min $ at the top of page 6. Seems to be consistent with definition of $ h^{*} $.
  - The double superscript of $ \pm $ of $ \mathbf p $ in Prop 6 is confusing
  - Overload on the symbol $ \sigma $ for std.
  - Text of footnote 2 is missing. I assume this is to do with choosing sex instead of age for German Credit?
  - The stated generalized $ \alpha $-mean seems to actually be just the $ L_p $-norm. A citation of where the definition of "generalized $ \alpha $-mean" comes from would be useful as there seems to be other definitions which give different definitions (i.e., see "Amari, Shun-ichi. Information geometry and its applications" where its definitions gives $ \alpha = 2 $)

**Summary Of The Paper:**

The paper provides an analysis of fairness via regularizing a loss function with a suitably selected fairness measure / metric. In particular, the paper analysis this under a PAC learning setting and a sample limit (asymptotic) setting. A majority of the paper focuses on the PAC setting, where Rademacher complexity bounds are made and analysed. Some experiments are included.

**Summary Of The Review:**

Although the paper explores an important gap in looking at fairness in the PAC setting (with generalization), I recommend a reject given my listed concerns about novelty. In particular, the proof techniques are not novel and the results of Theorem 2 does not seem to extend the prior work of Agrawal et al. (2020). Furthermore, the experiments do not seem well connected to the theoretical results explored [1]. The  notation choices, typos, and missing definitions further cements that the paper needs major revisions.

[1] Personally, I would even recommend excluding the experiments in the main text for further analysis, i.e., how Rademacher complexity terms might change when considering different fairness criteria; or connections of $ Z_{S} $ to known fairness quantities like data representation rate.

---

### Official Review · Reviewer_bVZW · 2021-11-02

**Correctness:** 4
**Technical Novelty And Significance:** 2
**Empirical Novelty And Significance:** 2
**Recommendation:** 3
**Confidence:** 4

**Main Review:**

Pros:

- Understanding the generalization properties of popular fairness measures, as well as their interplay with accuracy, is certainly an interesting problem.

- Sections 2 and 3 are quite well-written, easy to follow and provide some intuitive results.

Cons:

Unfortunately, I find the rest of the paper very hard to follow. This is especially due to two reasons:

- It is not clearly stated what the purpose/goal of the paper is. In particular, results follow one after the other, without much connection between them. I would suggest that the introduction is thoroughly rewritten, to clarify what the goal of this work is and how the results that follow address the problem at hand.

- The comparison to related work is kept very vague. As the author acknowledge, there are multiple previous studies on the generalization properties of fairness notions. It is claimed in the text that the present work generalizes these previous results, but it is not clarified in what way.



**Summary Of The Paper:**

The paper studies the generalization properties of PAC learning with fairness constraints.

**Summary Of The Review:**

While the topic of the paper is interesting and some of the results look promising, I do not recommend acceptance due to the lack of clarity about the objectives of this work and the insufficient comparison to prior work.

---

### Official Review · Reviewer_myf3 · 2021-11-02

**Correctness:** 4
**Technical Novelty And Significance:** 2
**Empirical Novelty And Significance:** 2
**Recommendation:** 5
**Confidence:** 2

**Main Review:**

In this paper, intuitive notations are used so that readers can easily follow, and theorems and proofs are well organized. The authors derive theoretical results by combining the generalization bound for standard statistical loss and the bound for fairness constraints. In the proofs of theoretical results, the process seems quite standard. It is interesting to observe that the portion of the sample for each group has a different way of affecting the standard loss term and the fairness term. And the two take-aways from proposition 6 are also interesting.

Some suggestions :
* Below equations (1) and (2), it is written as $T=D, S$, which would be nice to write as $T\in\{D,S\}$ for consistency.
* Missing part in two $h_{T}^{*}={ }_{h \in \mathcal{H}} \mathcal{L}_{T}(h)$.
* Section 4.2 mentioned that probabilistic upper bound is useful as a practical check, but it does not seem to reach me practically yet. To emphasize this more, it would be nice to mention the tightness of the upper bound of eq (8), or to add simple experimental results checking it.
* I think it would be nice to show something more experimentally. Analysis on binary classification was performed, but non-binary classification results can also be included in experiments. And, analysis is with non-binary groups, so experiments can be performed on that setting. Also, you can consider showing some experimental results aligning with proposition 6.

**Summary Of The Paper:**

This paper establishes the guarantee for the generalization of fairness-aware learning in binary classification under PAC-learning and a more practical asymptotic framework. Through the derived theorem, authors conclude that low sample size and class balance lead to the poor generalization of fairness-aware learning, and the need for a sample-efficient method is also argued. The experimental results using real data are aligned with the theorem, emphasizing the validity of theorem.

**Summary Of The Review:**

It is meaningful to establish a guarantee for generalization in fairness-aware learning, but theorem and proof processes are too standard, so novelty seems to be insufficient. Considering practice, additional theoretical results or more diverse experiments expected to be added.

---

### Official Review · Reviewer_CZ3f · 2021-11-03

**Correctness:** 2
**Technical Novelty And Significance:** 2
**Empirical Novelty And Significance:** 2
**Recommendation:** 3
**Confidence:** 4

**Main Review:**

The paper is hard to parse from time to time, and the organization of material is more or less confusing.

### q1: "this paper is first and foremost concerned with ... [tackling] issue of generalization"

To the best of my knowledge, previous literature contains (at least some) results regarding the generalization bound (for example, Woodworth et al. (2017) (c.f., Theorem 7)). If there is any misunderstanding please kindly let me know.

### q2: "the derivation of PAC framework for fairness trade-offs"

- In Table 1 the authors list several fairness notions, I am just wondering if the proposed bounds apply to all notions? If so, there is a worry of the tightness and the practical significance of the bound; if not, which notions could fit in the proposed framework?

- In Proposition 3, how strong is the assumption on $\phi$? It would be very helpful if the author can connect the technical detail with the fairness notion of interest, or provide some insights regarding how to efficiently check if one can directly apply the presented result.

### q3: the derivation of the asymptotic regime

In Section 5, Theorem 2 presented a convergence in distribution. While I can understand the composition of the variance term (as noted in the paper), I am not sure how to parse the result. What is the purpose of deriving this convergence in distribution?

**Summary Of The Paper:**

The paper claims to study out-of-sample generalization w/ both (unconstrained) loss and the fairness consideration. While there are multiple theoretical results presented, the connection between them could be further elaborated, so that the takeaway messages can be clearly conveyed.

**Summary Of The Review:**

Overall, the paper is relatively hard to follow, and it would be greatly appreciated if the authors can kindly clarify the questions/concerns as detailed in `Main Review`.

---

### Decision · Program_Chairs · 2022-01-20

**Decision:**

Reject

**Comment:**

This paper establishes the guarantee for the generalization of fairness-aware learning in binary classification under PAC-learning and a more practical asymptotic framework. The paper is nicely written, and theorems and proofs are well organized. However, novelty of the contribution seems to be insufficient. A future version of the paper may benefit from additional theoretical results or more diverse experiments.